# Analysis of single-cell transcriptomes links enrichment of olfactory receptors with cancer cell differentiation status and prognosis

Siddhant Kalra[1,7], Aayushi Mittal[1,7], Krishan Gupta [1,2], Vrinda Singhal[1], Anku Gupta[2], Tripti Mishra[3], Srivatsava Naidu[4], Debarka Sengupta [1,2,5,6✉] & Gaurav Ahuja [1✉]

Ectopically expressed olfactory receptors (ORs) have been linked with multiple clinically-relevant physiological processes. Previously used tissue-level expression estimation largely shadowed the potential role of ORs due to their overall low expression levels. Even after the introduction of the single-cell transcriptomics, a comprehensive delineation of expression dynamics of ORs in tumors remained unexplored. Our targeted investigation into single malignant cells revealed a complex landscape of combinatorial OR expression events. We observed differentiation-dependent decline in expressed OR counts per cell as well as their expression intensities in malignant cells. Further, we constructed expression signatures based on a large spectrum of ORs and tracked their enrichment in bulk expression profiles of tumor samples from The Cancer Genome Atlas (TCGA). TCGA tumor samples stratified based on OR-centric signatures exhibited divergent survival probabilities. In summary, our comprehensive analysis positions ORs at the cross-road of tumor cell differentiation status and cancer prognosis.

[1] Department of Computational Biology, Indraprastha Institute of Information Technology-Delhi (IIIT-Delhi), Okhla, Phase III, New Delhi 110020, India. [2] Department of Computer Science and Engineering, Indraprastha Institute of Information Technology-Delhi (IIIT-Delhi), Okhla, Phase III, New Delhi 110020, India. [3] Pathfinder Research and Training Foundation, 30/7 and 8, Knowledge Park III, Greater Noida, Uttar Pradesh 201308, India. [4] Center for Biomedical Engineering, Indian Institute of Technology Ropar, Bara Phool, Birla Seed Farms, Rupnagar, Punjab 140001, India. [5] Centre for Artificial Intelligence, Indraprastha Institute of Information Technology, Okhla Phase III, New Delhi 110020, India. [6] Institute of Health and Biomedical Innovation, Queensland University of Technology, Brisbane, Australia. [7] These authors contributed equally: Siddhant Kalra, Aayushi Mittal. ✉email: debarka@iiitd.ac.in; gaurav.ahuja@iiitd.ac.in

Sensory inputs play a vital role in many essential behaviors, ranging from the identification of prey or peer mates to the incoming danger[1–4]. Chemoreception, in particular, is a key sensory system, which is largely mediated by olfactory and taste receptors[5–7]. In the human genome, olfactory receptors (ORs) constitute the largest gene family with ~400 functional and ~600 non-functional pseudogenes[8]. These functional olfactory receptors are further classified into distinct subfamilies. In mammals, the olfactory receptor families include odorant receptor (ORs)[9], vomeronasal type 1 and type 2 receptors (V1Rs and V2Rs)[10], trace amine-associated receptors (TAARs)[11], formyl peptide receptors (FPRs)[12], and the membrane guanylyl cyclase (GC-D)[13]. Based on their evolutionary relationship, ORs, the largest gene sub-family, are further classified into Class I and Class II receptors[9]. Vomeronasal receptors (VRs), also known as pheromone receptors, are sub-classified into V1Rs and V2Rs[10]. TAARs, another subfamily of chemosensory receptors, are known to detect trace amines such as β-phenylethylamine, p-tyramine, tryptamine, and octopamine[14]. Membrane-associated Guanylyl cyclase receptors have been shown to be activated by membrane diffusible nitric oxide and other ligands such as uroguanylin and guanylin[15].

The olfactory signal transduction initiates with the binding of odorants to the olfactory receptors, which in turn triggers the $G_{alpha}$ protein-mediated activation of adenylate cyclase, leading to an increase in cyclic adenosine monophosphate (cAMP) levels[16]. Elevated cAMP levels interact and instigate the opening of cyclic nucleotide-gated (CNG) channels, resulting in the influx of cations, mainly sodium ($Na^+$) and calcium ($Ca^{2+}$) ions, ultimately leading to the depolarization of olfactory sensory neurons (OSNs). This transient increase of the intracellular $Ca^{2+}$ ions triggers the opening of $Ca^{2+}$-activated chloride (CaCCs) channels that amplify the CNG channel signal[16].

Notably, in addition to their expression in the OSNs of the olfactory epithelium, ORs exhibit ectopic expression in non-olfactory tissues such as muscle, kidney, and keratinocytes[17–22]. Advancements in the high-throughput sequencing technologies accelerated the systematic exploration of the ectopic chemosensory receptors in almost all human tissues[19]. In addition to this, OR-expression has also been reported in epithelial malignancies[23–25]. OR expression in non-olfactory tissues has been found to play a crucial role in various physio-molecular processes including wound healing, cellular motility, sperm chemotaxis, and the regeneration of muscle cells[20,26–28]. Moreover, various functional studies highlighted the therapeutic promise of ligand-mediated OR activation in cancer[19,24]. For instance, in the case of hepatocellular carcinoma, the activation of OR1A2 by citronellal leads to a cAMP-dependent increase in cytosolic $Ca^{2+}$ ions, thereby impeding cancer cell proliferation[29]. Similarly, Troenan induced activation of OR51B4 in colorectal cancer cells inhibits their migratory potential and instigates apoptosis[30]. Moreover, in prostate cancer, activation of OR51E2 by an endogenous agonist, 19-hydroxyandrostenedione resulted in neuroendocrine trans-differentiation, revealing the functional implication of ORs in diverse physio-molecular processes[31]. In addition to this, in vitro activation of OR51E2 in vertical-growth phase melanoma cells by β-ionone resulted in the activation of anti-proliferative, anti-migratory, and pro-apoptotic pathways[32]. Similar findings have been reported for non-small-cell lung cancer where OR2J3 activation by helional leads to the induction of apoptosis and anti-proliferative pathways[33]. Notably, due to their expression specificity in the tumor-state, a handful of ORs have been identified as potential biomarkers e.g. OR51E1 in small intestine neuroendocrine carcinomas[34], OR7C1 in colorectal cancer[35], PSGR in prostate cancer[36], and OR2B6 in breast carcinomas[23]. All these collectively reinforce the idea of the potential therapeutic and diagnostic function of these ectopic olfactory receptors.

Traditional bulk-tissue based transcriptomics estimates the average expression of individual ORs across admixture of cell types within the highly heterogeneous tumor biopsies, leading to under-detection of lowly or selectively expressed ORs[37,38]. In the present study, we investigated the expression profiles of ORs in single malignant cells leveraging numerous publicly available single-cell RNA sequencing (scRNA-Seq) datasets. We could identify 59 previously unreported OR-tumor pairs. Our results indicate that across cancer types, intra-tumoral heterogeneity concurs with the number of expressed ORs per cell. Delineation of the breast epithelial malignant cells along the cellular differentiation trajectory revealed a substantial decrease in the cellular count of expressed ORs and their expression, along with the differentiation kinetics. Finally, we were able to construct OR-centric transcriptomic signatures, which stratifies breast cancer samples from TCGA into distinct groups with divergent survival probabilities.

## Results

**Widespread expression of OR genes in malignant cells**. Activation and upregulation of the olfactory receptor genes in various cancer types are well established, however, in most cases, the assessment of the OR transcripts is achieved by profiling of the bulk tumor samples, which largely masks both the OR expression as well as information about the contributing cell-types, thereby obscuring their adaptation in diagnosis and management of cancers[38]. To address this, we developed a computational workflow (CancerSmell) that systematically estimates the activation status of the chemosensory olfactory and taste receptors at the single-cell resolution (Supplementary Fig. 1a). Using this, we evaluated the expression of these chemosensory receptors across 49 scRNA-seq datasets (tumor and cell lines), featuring 22 tumor types and collectively comprising 42,529 malignant cells (Supplementary Fig. 1b). Our workflow identified numerous chemoreceptor-tumor pairs which were largely unreported in cancer literature (Figs. 1a, 2a, Supplementary Fig. 2a, b, d–f). Notably, among them, we have observed enrichment of OR genes in cancer, in contrast to TAARs, V2Rs and taste receptors (T1Rs and T2Rs), which is in line with the earlier reports, where the comparative enrichment of ORs is reported in the cancer cells, in contrast to other chemosensory receptors (Fig. 1a, c–f, Supplementary Fig. 2c, d–f; Supplementary Data 1). Next, we sought to determine the specificity and exclusivity of the identified ORs towards cancer-types. Further assessment of these results revealed two distinct classes of ORs, based on their tumor specificity. For example, OR8B8 and OR8H1 were exclusively detected in malignant breast epithelial cells, whereas OR1A1 and OR2M3 activation were observed in 11 and 10 different tumor types respectively, suggesting a distinct mode of expression regulation (Fig. 1a, Supplementary Fig. 2a, b; Supplementary Data 2). Of note, ORs with pseudo-gene status were vastly undetected across the studied expression datasets (Supplementary Fig. 2a, b). Next, we asked if tumor-associated ORs exhibit bias towards certain genomic loci. In our analysis, all the detected ORs were restricted to only 10 chromosomes, out of the 21 OR coding chromosomes[39], with chromosome 11 harboring the largest fraction of these receptors (Fig. 1b). We observed a high degree of variability in their expression and cellular detection frequency, alluding to their potential contribution in tumor heterogeneity (Fig. 1e, f). Functional studies elucidating the role of these tumor-associated olfactory receptors indicate the involvement of these receptors in regulating key cancer-related pathways[40]. For this purpose, we used the Gene Set Variation Analysis (GSVA) library. Our results indicate that a significant proportion of the detected ORs are implicated in processes such as stemness, metastasis, invasion and differentiation (Fig. 2b–e, Supplementary Fig. 2g, Supplementary Data 3).

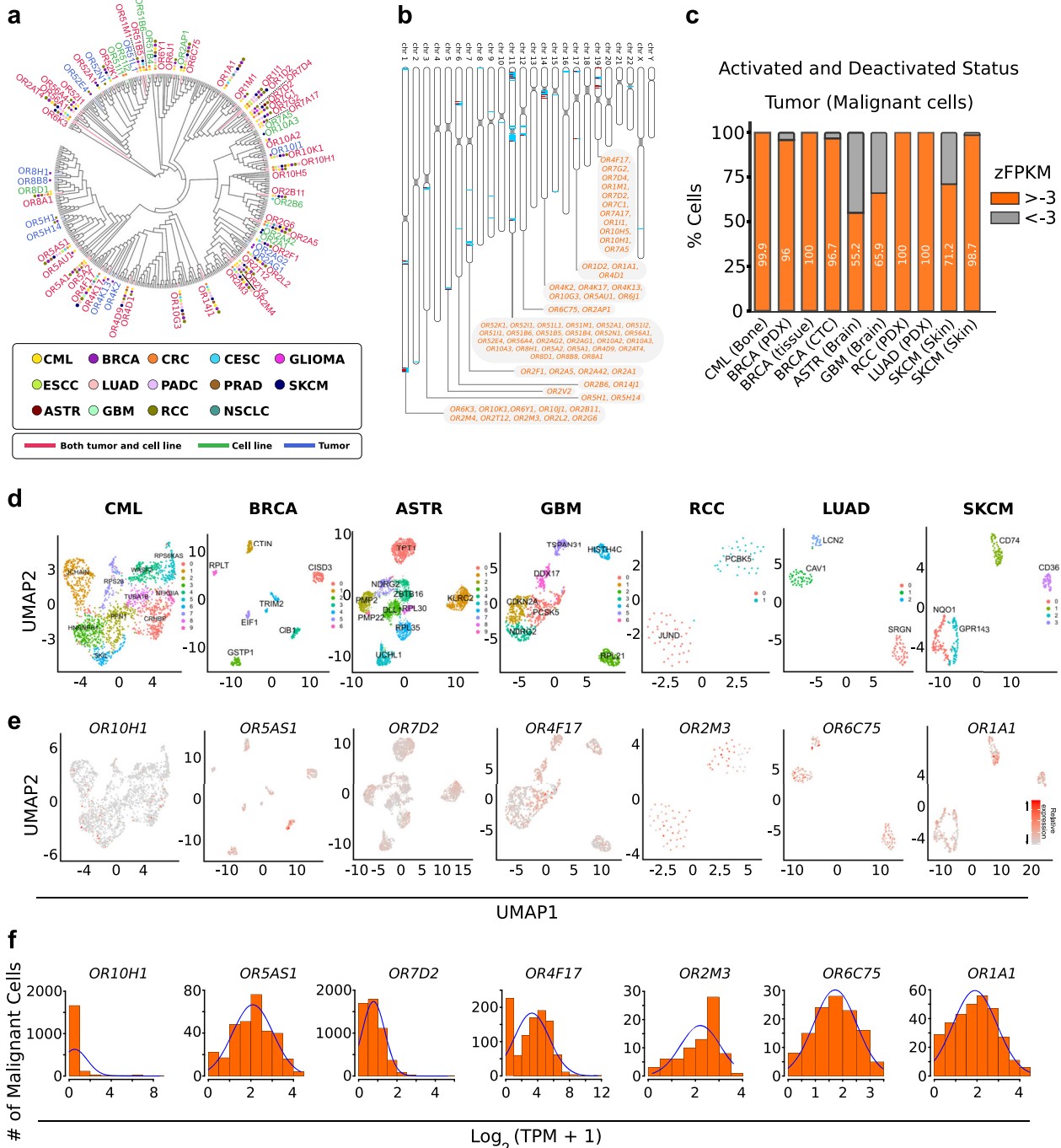

**Fig. 1 Comprehensive catalog of tumor-associated ORs at the single-cell resolution. a** Phylogenetic tree representing ORs sequence relationships and their detection status in 14 distinct cancer types. The tree was constructed using the PhyML algorithm. The protein sequences of the functional ORs were aligned using the MUSCLE sequence alignment algorithm. Colored circles represent the tumor types and the branch colors represent the source information. The green, purple, and red branch color represent the presence of the indicated OR in the cell line, tumor, or both respectively. **b** Ideogram illustrating the chromosomal location of all the functional ORs in humans. Red lines represent the genomic loci of the functional ORs reliably detected in the malignant cells (tumor or cell lines), whereas cyan lines represent the genomic loci of ORs that are not detected in any tumor-types or cell lines under investigation. The identities of the malignant cells-associated ORs are indicated in the box. **c** Bar graph representing the percentage of OR-positive malignant cells in the indicated ten different tumor single-cell datasets. zFPKM algorithm was used for the determination of the OR activation status (zFPKM >−3, activated). The percentage of active cells (OR-positive) represents the proportion of cells possessing zFPKM values >−3 for any functional OR. **d** Uniform Manifold Approximation and Projection (UMAP) based embedding of single-cell expression profiles representing the distinct cell types in the indicated tumor datasets. Different clusters are depicted with distinct colors. Cells within a cluster represent similar cell types at the transcriptome level. Markers for each cluster are indicated as text. These markers were identified by using the "FindAllMarkers" function of the Seurat (v 3.1.1). **e** Uniform Manifold Approximation and Projection (UMAPs) depicting the relative expression of the representative ORs in the indicated single-cell tumor datasets. Cells within a cluster represent similar cell types. **f** Density-histograms depicting the normalized expression of the indicated ORs in the corresponding tumor type. Y-axis represents the number of malignant cells expressing the indicated OR whereas the X-axis denotes the normalized expression value as $\log_2(TPM+1)$.

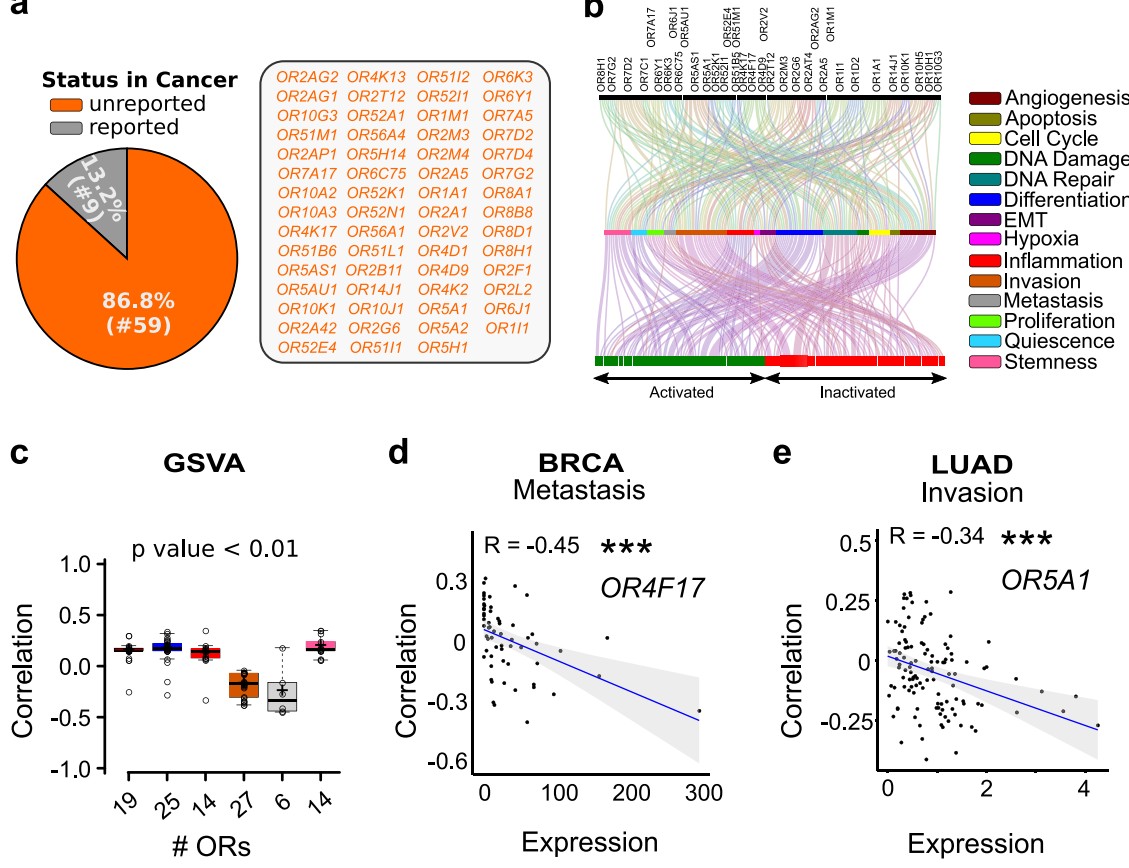

**Fig. 2 Functional enrichment analysis of tumor-associated ORs revealed their potential role in regulating key cancer-related pathways. a** Estimation of the olfactory repertoire at the single-cell resolution revealed previous unreported cancer-associated olfactory receptors, represented here as a Pie Chart. CancerSmell computational workflow identified 59 previously unreported malignant cells-associated ORs. **b** Gene Set Variation Analysis (GSVA) revealed the functional relevance of the indicated olfactory receptors in tumor-related processes, indicated here as the Alluvial plot. The activated (GSVA score > 0 and corrected *p*-value < 0.01) and inactivated (GSVA score <0 and corrected *p*-value < 0.01) status of the indicated biological pathways is represented along the horizontal bar. **c** Box plots depicting the correlation between GSVA scores and the OR expression in the indicated tumor-related signatures. The *X*-axis represents the total number of ORs identified to be regulating the indicated biological process. The color of the box represents distinct tumor signatures (mentioned in Panel **b** of Fig. 2). **d** Scatter plots representing the correlation between GSVA scores for metastasis signature and OR4F17 in the breast carcinoma cells. The *R-value* designates the correlation coefficient, whereas the *p*-value indicates the statistical significance. **e** Scatter plots representing the correlation between GSVA scores for invasion signature and OR5A1 in the lung adenocarcinoma cells. The *R-value* designates the correlation coefficient, whereas the *p*-value indicates the statistical significance.

**Aberration of the one-receptor one-cell rule in cancer**. OR gene expression is tightly regulated in the OSNs, leading to the expression of a single OR gene in a mature neuron[41,42]. We sought to determine if similar transcriptional regulation is also applicable during ectopic OR expression in the malignant cells. To test this, we first estimated the cellular frequencies of co-expressed ORs across scRNA-seq datasets of numerous tumors or cell lines. Our results suggest that unlike mature OSNs, a significant proportion of the malignant cells express multiple ORs, except in the case of tumors related to the nervous system i.e. glioblastoma and astrocytoma, where we found malignant cells to obey the "one or none" rule (Fig. 3a, Supplementary Fig. 3a). Next, we asked whether the co-expression of ORs in tumors is tightly regulated or stochastic in nature. Our results indicate that the selection of the expressed ORs within a single cell is stochastic and is poorly correlated to their expression (BRCA dataset; R = 0.34; *p*-value < 0.0001) (Fig. 3b, c, Supplementary Fig. 3g, h). Past reports evaluating the expression of chemosensory receptors in healthy tissues revealed the presence of multiple ORs[23,43]. We, therefore, examined the exclusivity of the ORs expressed in tumors. Comparative analysis revealed that the number of activated OR genes are systematically higher in

the malignant state. Only a sub-fraction of these (14 out of 53 BRCA-associated ORs) were detected in healthy cells (Fig. 3d–f; Supplementary Fig. 3b–d; Supplementary Data 4), which is largely in line with the previous reports[19]. Among all the studied cancer types, breast malignancies portrayed the maximum relative abundance of OR transcripts, inspiring us to closely pursue the concerned cancer type. We, therefore, estimated the propensity of expressing multiple ORs per cell in the distinct molecular subtypes of breast cancer. Noteworthily Triple-Negative Breast Cancer (TNBC) cells, being the most aggressive breast cancer subtype[44], displayed the highest levels of enrichment of OR repertoire (Supplementary Fig. 3e; Supplementary Data 5). Moreover, we observed no significant variations in breast carcinoma-associated ORs expression across the molecular subtypes (Bonferroni corrected *P*-value > 0.05) (Supplementary Fig. 3f). Further, we inspected the coherence of OR expression with the well-known, cancer-related functional states. We observed an inverse relationship between OR enrichment and pro-tumor signatures such as invasion, metastasis, proliferation, and DNA damage (Fig. 3f). Notably, for some ORs, a positive correlation was observed between their expression and signatures related to stemness, differentiation, and angiogenesis,

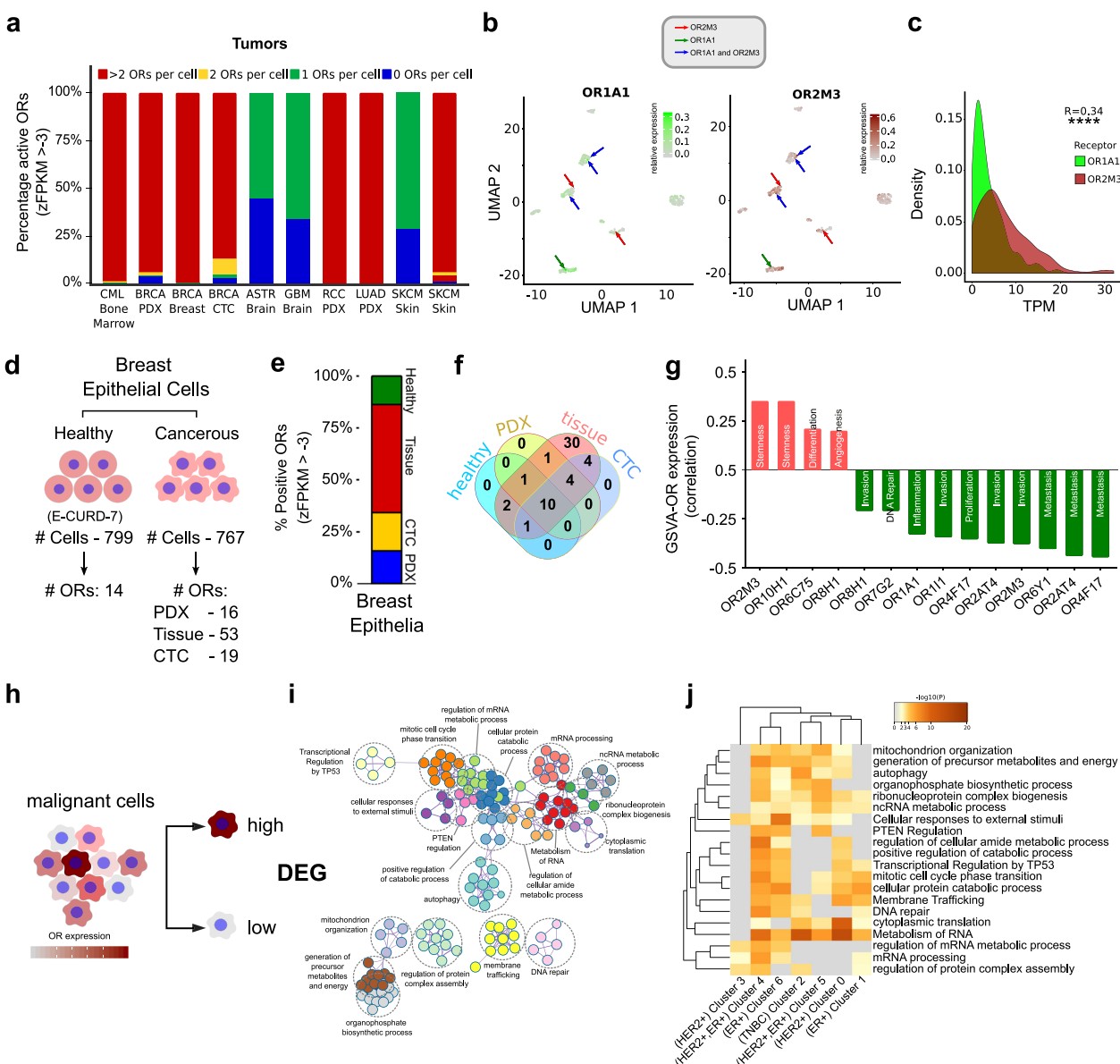

**Fig. 3 Cancer cells express multiple olfactory receptors. a** Cellular count of expressed ORs largely varies across multiple tumor types, depicted here as a percentage bar graph in the indicated tumor-types. zFPKM algorithm was used for the determination of the OR activation status (zFPKM >−3, activated). **b** Uniform Manifold Approximation and Projection (UMAPs) representation of the cellular expression of two representatives ORs in the breast carcinoma single-cell dataset. The red-colored arrows indicate the OR2M3 expressing malignant cells, whereas the green arrow denotes the OR1A1 expressing malignant cells. Notably, the cells indicated via blue arrows co-express both of these receptors. The scale bar on the right represents the relative expression values of the indicated ORs. **c** Density plot depicting the expression variability between the indicated ORs in the breast carcinoma single-cell dataset. The p-value significance and the correlation coefficient is depicted on the right. **d** Graphical illustration depicting the total number of single cells and the reliably detected ORs in the healthy and malignant breast epithelial cells. **e** Percentage bar graph depicting the relative proportion of detected ORs in the indicated healthy and malignant epithelial cells. The different conditions (healthy, tissue, CTC, and PDX) are indicated by different colors. **f** Venn diagram depicting the number of overlapping ORs in the indicated conditions. **g** Bar graph depicting the correlation between GSVA scores of the indicated biological process and ORs expression across all cells. Notably, the positive and negative correlated values are indicated in red and green colored bars, respectively. **h** Schematic representation depicting the strategy employed for differential gene expression analysis. Notably, the malignant cells were segregated into two subcategories based on the expression of ORs per cell. Differentially expressed genes were calculated using the Wilcox test. **i** Metascape analysis of differentially expressed genes depicting the functional importance of BRCA-associated ORs in the highlighted biological/molecular processes. **j** Heatmap depicting cluster-wise enrichment of the prominent biological functions. Scale bar represents the negatively log-transformed (base 10) p-values.

implying their contrasting implications in the activation of cancer-related biological pathways (Fig. 3g). Past studies (both in vivo or in vitro models*)* have linked ligand-mediated OR activation with multiple non-canonical molecular processes. To this end, we segregated the single-cell malignant breast epithelial cells based on the overall enrichment of expressed OR genes and

functionally annotated the differential genes between the concerned cell-groups (Fig. 3h, Supplementary Data 6). Key molecular processes thus retrieved, included regulation of cell cycle, transcriptional or translational regulation, autophagy, etc. (Fig. 3i, j, Supplementary Fig. 3i). To summarize, our results suggest that cellular count of expressed ORs and their respective

expression levels concur with clonal heterogeneity in breast tumors, both at the molecular and functional levels.

**Correlation between ORs and tumor differentiation status.** Recent reports studying mammals suggest the involvement of a special epigenetic mechanism behind the simultaneous expression of multiple ORs in developmentally immature neurons[41,45]. Cancer cells also undergo extensive epigenetic reprogramming such as DNA methylation, promoter hypermethylation, change in chromatin structure, and histone modifications during intratumor cellular (de)differentiation[46]. Such changes are known to be responsible for the dysregulation of the cell cycle control, thereby affecting cell proliferation and survival mechanisms of the cell. To this end, we tracked the change in the cellular count of expressed ORs in the context of cancer-cell differentiation. For this, we performed pseudo-time based reconstruction of the differentiation trajectories underpinning clonal expansion of malignant breast epithelial cells[47]. Notably, the breast cancer single-cell dataset consists of single-cell expression profiles of 11 treatment-naive and 1 under-treatment patients (Supplementary Data 7). Our initial analysis was performed on cells from all molecular subtypes, except the cells from patient BC05, who reportedly underwent neoadjuvant immunotherapy. Monocle yielded three main branches elucidating the emergence of differential pathological stages entailing molecular subtypes (Supplementary Fig. 4a–d). Notably, for this analysis, we manually selected the starting-point of the inferred trajectory based on the low-dimensional spatial agglomeration of cells harboring maximum

relative stemness scores (Supplementary Fig. 4c). A negative correlation was observed between pseudotime and cellular stemness ($R_{stemness} = -0.27$, $p$ value = <0.0001) (Supplementary Fig. 4e). In contrast, minor ($R = 0.2$) or no significant correlation ($R = -0.021$, $p$-value = 0.75) was observed between cellular OR expression or the number of expressed ORs per cell along the pseudotime, respectively (Supplementary Fig. 4f, g). Upon closer inspection of the individual subpopulation of Luminal cells, it offered a strong negative correlation between pseudotime and cellular count of expressed ORs ($R_{ORfrequency} = -0.85$, $p$ value = <0.0001) (Fig. 4a–c, Supplementary Fig. 4i). Moreover, similar results were obtained for cellular stemness along the pseudotime ($R_{stemness} = -0.77$, $p$ value = <0.0001) (Supplementary Fig. 4h). Conversely, we have observed a strong positive correlation between cellular stemness and its expressed OR repertoire ($R = 0.55$, $p$ value = <0.001) (Supplementary Fig. 4j). Next, we asked whether such a steep decline in the cellular count of expressed ORs or their expression along the cellular differentiation trajectory is specific to malignancy. To test this, we have conducted a similar analysis with the healthy luminal breast epithelial cells which revealed no significant anti-correlation ($R = 0.048$, $p$ value = 0.4) (Fig. 4d–f, Supplementary Fig. 4k–n; Supplementary Data 7).

**ORs-centric signatures stratify BRCA tumors.** Tumor environments harbor multiple cell-types, thereby obscuring the detectability of cancer-specific ORs through bulk transcriptomics. In breast cancer, the differentiation stage is used as a parameter

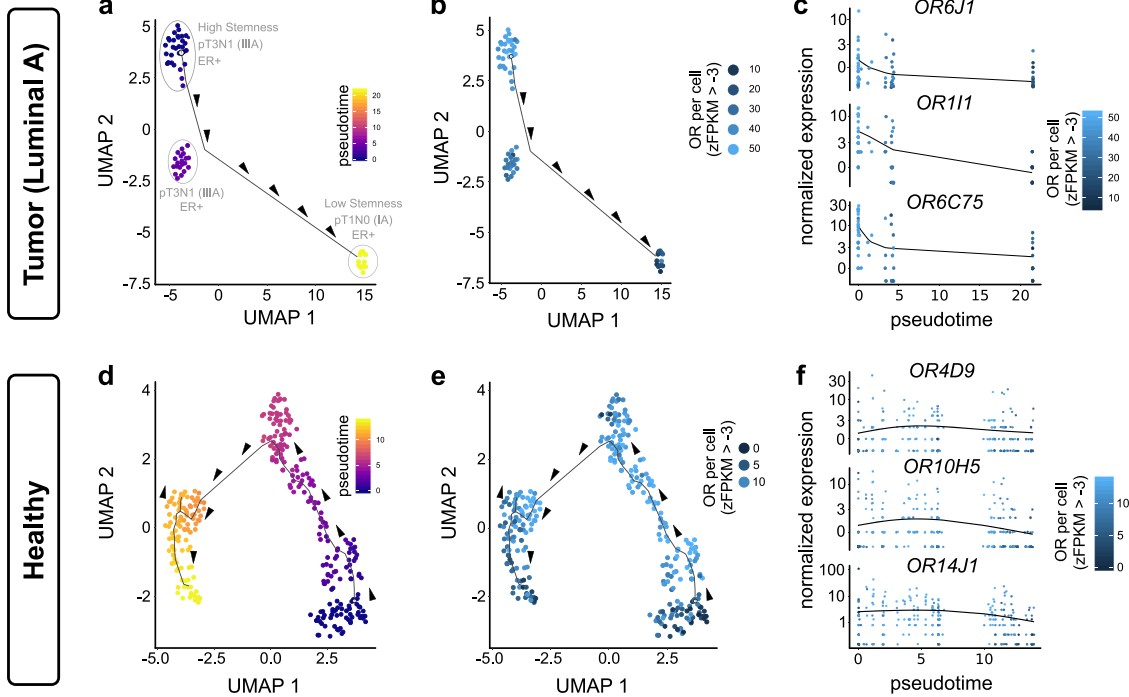

**Fig. 4 Cellular decrease of expressed ORs with higher differentiation status. a** Monocle-generated pseudotemporal trajectory of malignant breast epithelial cells of luminal A molecular subtype of breast carcinoma, depicting the decrease in cellular stemness during cellular differentiation time-course (pseudotime). The cells with high stemness properties were selected as a starting point. Arrowheads indicate the direction of cellular differentiation across the pseudotime. Scale bar represents the pseudotime. **b** Uniform Manifold Approximation and Projection (UMAP) represents the decrease in the number of expressed OR genes per malignant cell in luminal A subtype in breast carcinoma during pseudotemporal trajectory. Scale bar represents the number of expressed OR genes per cell. Arrowheads indicate the direction of cellular differentiation across the pseudotime. **c** Scatter plots depicting an overall decrease in the expression of representative ORs along the pseudotemporal trajectory in the indicated conditions. Scale bar represents the number of expressed OR genes per cell. **d** Pseudotemporal trajectory of healthy luminal breast epithelial cells. Arrowheads indicate the direction of cellular differentiation. Scale bar represents the pseudotime. **e** Uniform Manifold Approximation and Projection (UMAP) representing the number of expressed OR genes per cell in the healthy luminal breast epithelial cells along the pseudotemporal trajectory. **f** Scatter plots depicting the expression dynamics of representative ORs along the pseudotemporal trajectory in the indicated conditions.

for the histopathological grading of tumor samples. To date, the clinical assessment and prognosis are largely done using The Nottingham Grading System, which incorporates the differentiation status of the tumor cells[48]. In furtherance to our observation about the overall decline in the cellular OR number/ expression associated with differentiation, we asked if multivariate OR signatures can be used to deconvolute the bulk expression profiles and could segregate patients with distinct survival. To test this, we first constructed the OR-centric signature capturing the co-activation of the expressed ORs (Supplementary Fig. 5a–c; Supplementary Data 8). Next, we projected these OR-centric signatures on TCGA bulk-transcriptomic breast carcinoma data[49], annotated with survival information, and obtained multiple patient groups, with at least two major groups having significantly distinct survival probabilities (Fig. 5a–c). Notably, we obtained consistent results from signatures constructed from three independent and biologically distinct breast cancer single-cell datasets, suggesting the robustness of the OR-centric signatures in tumor classification. Further dissection of this multivariate analysis revealed that the patient group possessing higher cosine distance with the OR-centric signatures representing the low number of expressed OR genes per cell harbors a good prognosis and better survival, which largely unmatches with estimated tumor stage of the patients (Fig. 5d–f, Supplementary Fig. 5d–f; Supplementary Fig. 6a–c). Taken together, these results reinforce the potential clinical relevance of tumor-associated ORs and demonstrate the prognostic value of the cell-type-specific, OR-centric signatures inferred from single-cell transcriptomes.

## Discussion

Dysregulation of the core transcriptional regulatory mechanisms in cancer results in the ectopic expression of normally silent genes[50,51]. One such example of the silent gene family is olfactory receptors. In addition to their expression in the sensory epithelium of the nose, a large proportion of the ORs are reported to be expressed in non-olfactory tissues, both under homeostatic as well as in pathological states such as cancer. Notably, both the expression as well as the number of expressed ORs highly increases in the malignant states (reviewed in ref. [19]). Although functional analysis with a handful of ORs has shown promising results in cancer diagnostics and therapeutics[19], still a comprehensive understanding of their presence in multiple tumor-types, their functional role in tumorigenesis, and ultimately in the tumor prognosis remains unclear. The present study addresses this gap by tracking OR expression in single malignant cells entailing 22 cancer types (tissues and cell lines) and 42,529 malignant cells. We reported 68 tumor-OR pairs, among which only ~15% have been mentioned in cancer literature. To the best of our knowledge, this is the first comprehensive single-cell study presenting a comprehensive analysis of the ectopically expressed

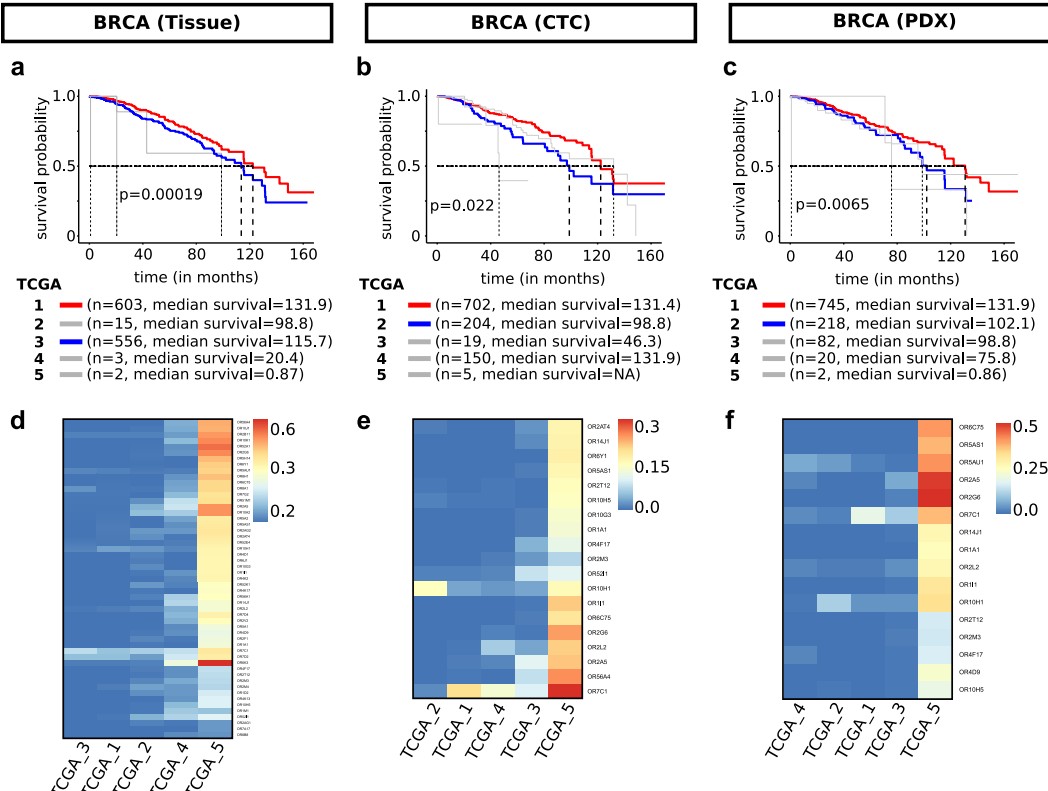

**Fig. 5 Single-cell transcriptome signatures stratify breast tumors into subgroups with distinct patient survival. a** Kaplan–Meier plot depicting the patient's survival in the indicated groups (indicated with colors) segregated based on the cosine similarity between OR-enrichment signatures, inferred from tissue-derived scRNA-sequencing profiling of malignant cells. **b** Kaplan–Meier plot depicting the patient's survival in the indicated groups (indicated with colors) segregated based on the cosine similarity towards OR-enrichment signatures, inferred from single-cell transcriptional profiling of circulating tumor cells. **c** Kaplan–Meier plot depicting the patient's survival in the indicated groups (indicated with colors) segregated based on the cosine similarity towards OR-enrichment signatures, inferred from transcriptomic profiling of breast carcinoma patient-derived xenograft-derived dataset. **d** Heatmap depicting the relative enrichment of breast tumor-derived ORs in the indicated subgroups of TCGA patients of breast carcinoma. **e** Heatmap depicting the relative enrichment of breast cancer circulating tumor cells-derived ORs in the indicated subgroups of TCGA patients of breast carcinoma. **f** Heatmap depicting the relative enrichment of breast tumor PDX-derived ORs in the indicated subgroups of TCGA patients of breast carcinoma.

olfactory and taste chemosensory receptors (functional ORs, pseudo OR genes, TAARs, V2Rs, taste receptor type 1 (T1Rs), and taste receptor type 2 (T2Rs) across multiple cancer-types. Molecular and functional analysis of the breast carcinoma-associated ORs alluded to their potential role in tumor cell (de) differentiation. Moreover, assessment of the DNA aberrations in the tumor-associated OR genes in breast carcinoma indicated DNA amplification events, which could be one of the underlying reasons for their upregulated expression in cancer (Supplementary Fig. 6d–f). In summary, we took advantage of the single-cell transcriptomics datasets and derived OR-centric signatures which could robustly stratify BRCA bulk tumor samples from TCGA with divergent survival probabilities.

Ectopically expressed ORs were first reported in the mammalian germ cells[52] and soon after, a multitude of effort was invested to uncover their expression across multiple human tissues. The availability of the high throughput next-generation transcriptome approaches, such as bulk RNA sequencing and microarray further accelerated this search and till now the expression of multiple ectopic ORs is reported in various human tissues under the homeostatic condition[18,53,54]. Interestingly, in addition to their reported expression in the healthy tissues, the past literature also offers a number of independent anecdotes linking elevated expression of specific ORs with the molecular prognosis of specific cancers[23,24,27,33,36,43]. Such examples include OR51B5 and OR2AT4 in leukemia[55], OR7C1 in colorectal cancer[35], OR51E2 in melanoma[32], OR10H1 in bladder cancer[56], OR51E2 in prostate cancer[31], OR51E1 in small intestine cancer[34], and OR2B6 in breast carcinoma[23,43]. Interestingly, the functional validations involving ligand-mediated activation of the aforementioned cancer-associated ORs resulted in a decrease in cancer cell proliferation or the complete cessation of the cancer growth, which further suggests their importance in clinical setups.

Notably, expression evaluation of these therapeutically relevant OR transcripts has been primarily achieved by targeted assays[57], which mask the information about the contributing cell-types and the expressing ORs. Single-cell transcriptomics circumvents these shortcomings by enabling investigation at the levels of individual cells, wherein cell-type identifiability remains tractable[58,59]. Added to it, single-cell expression allowed us to track the functionally interpretable combinatorial expression of ORs. Interestingly, the number of expressed ORs per cell systematically declined with luminal A cell differentiation, proving it to be a powerful molecular signature for cancer grading. Notably, such a negative correlation is exclusive to cell malignancy. Prognostic value of ORs was reinforced as we obtained distinct survival groups, simply by tracking the enrichment of key OR-centric molecular signatures in TCGA tumor samples. In this study, we make several observations shedding light on the potential relationship between widespread expression of a large spectrum of ORs and expansion/differentiation of cancer clones. Despite their presence in the malignant cells, so far the experimental validation of their predictive role in tumor-related molecular pathways is still elusive. Moreover, the majority of the identified ORs are orphan receptors, therefore, in order to delineate their contribution in tumor biology, identification of their agonist or antagonist is the first step forward. We assume that the cancer-specific metabolic intermediates could be the potential endogenous agonists for these cancer-specific ectopic ORs. Since the present results link the cellular count of expressed ORs with tumor cell differentiation, therefore, it is important to functionally validate these findings by the loss-of-function experiments. Notably, in the field of cancer biology, there are numerous well characterized phenotypic manifestations such as Epithelial-to-Mesenchymal transition[60], cancer cell stemness[61] and blockage of immune checkpoints[62] that are known to play a significant role in the disease progression and survival. Our results introduce ORs as a new dimension to the understanding of cellular differentiation and prognosis in cancer, which requires further investigations.

## Methods

**Computational workflow of Cancer Smell**. Cancer Smell computational workflow was used to identify the potential chemosensory receptor gene reliably expressed in single-cell RNA sequencing cancer datasets. It contains inbuilt information about human-specific chemosensory receptor repertoire, which was manually curated from public databases i.e. The Human Olfactory Data Explorer, Horde database (https://genome.weizmann.ac.il/horde/)[63] and Uniprot (https://www.uniprot.org). Cancer Smell takes a raw TPM matrix as input in which the genes and cell Id information are arranged in rows and columns, respectively. To ensure that the input data matrix represents single cells, Cancer Smell utilizes "scrublet" (v0.2.1)[64], a python package that was used on python v3.7.0. Notably, the cells which qualify the singlet criteria were used in the subsequent downstream analysis. Cancer Smell utilizes zFPKM (v1.8.0)[65], a Bioconductor package for the estimation of the gene activation status. It uses a recommended cutoff i.e. zFPKM value $> -3$ for the active genes[65]. For the downstream steps, Cancer Smell utilizes Seurat (v3.1.1)[66] for data scaling, normalization, dimension reduction, and clustering.

**Removal of doublet cells in scRNA-seq datasets**. This step is performed to selectively identify and filter concatenated cells in the scRNA-seq dataset. The python script based on scrublet (0.2.1)[64] (available on the GitHub link), takes a raw TPM or FPKM matrix as an input. The cell information is represented in rows whereas the columns contain gene names. Scrublet returns a boolean matrix with information about the doublet status of each cell. The doublet cells were filtered from the matrix and the remaining cells were subjected to further downstream analysis.

**Identification of gene activation status**. zFPKM approach was used to determine the set of genes that are functionally active and inactive in a particular cell[65]. By using the zFPKM package in R, the expression scores were converted into zFPKM scores and the median value of these scores for each gene was computed. The genes which possess the median value of the scores $> -3$ were considered as active.

**Classification of cell-types by Seurat software suite**. The downstream analysis for every single cell across a pan tumor was implemented using Seurat (v3.1.1)[66]. The filtered expression matrix is used as an input file for the downstream analysis. The inbuilt function "CreateSeuratObject" converts the input data file into a Seurat class R object. Further, the data was scaled and normalized and the principal components were identified using ScaleData and RunPCA functions respectively. KNN graph-based approach based on euclidean distance was used to cluster the data points for which "FindCluster" inbuilt function was used with its default parameters. Once the clusters were obtained, they were overlaid with the metadata to decipher meaningful biological information. Each cell was classified as positive and negative depending on the expression of chemosensory receptors within that cell.

**Pathway enrichment analysis using GSVA**. The functional state of the cell harboring chemosensory receptors was estimated using Gene Set Variation Analysis (GSVA; v1.34.0)[67], a non-parametric unsupervised method used for pathway enrichment for each sample. In this study, fourteen distinct tumor-related signatures were used, namely angiogenesis, apoptosis, cell cycle, differentiation, DNA damage, DNA repair, EMT, hypoxia, inflammation, invasion, metastasis, proliferation, quiescence, and stemness.

**Construction of phylogenetic tree**. SeaView software (v1:4.6.4-1) was used for constructing a sequence-based phylogenetic tree. FASTA sequences of functional ORs were downloaded from the Ensembl genome browser (BioMart) and were used to construct the unrooted phylogenetic tree. The sequences were aligned using MUSCLE (Multiple Sequence Comparison by Log Expectation) and the PhyML algorithm was used for tree construction. Implementation of additional graphical features was performed using an interactive tree of life (iTOL) (https://itol.embl.de/).

**Ideogram construction**. WashU Epigenome Browser (v46.2) with human genome version Hg38 as a reference genome was used for the ideogram construction. The two-color scheme was used to discriminate between cancer-associated (red) and non-cancer-associated ORs (blue).

**Functional enrichment analysis**. To identify the differentially expressed genes between the cells harboring high and low numbers of expressed ORs, we first calculated the median of the OR count across all cells. Next, we segregated the cell population (cluster-wise) based on the median value. Lastly, we calculated the differential gene expression analysis using the Wilcox-test. Only statistically significant genes were selected for the functional enrichment analysis (fold change cutoff $+-4$; $p$-value $< 0.05$) using Metascape (http://metascape.org/).

**Pseudo temporal Ordering of single cells using Monocle 3**. Cell fate decisions and differentiation trajectories were reconstructed with the Monocle 3 package. This computational workflow utilizes reverse graph embedding based on a user-defined gene list to generate a pseudotime plot that can account for both branched and linear differentiation processes. In order to understand how the changes in the OR expression or their cellular counts relate to breast cancer cell stemness, we initially computed the degree of stemness in each individual malignant cell for the determination of the starting point for the trajectory formation. The extent of stemness for a particular cell was calculated using the GSVA scores for Stemness, which collectively assigns a stemness score for each cell using known bonafide stem cell signatures. Ordering of the cells in the 2D space was achieved using an inbuilt function of Monocle 347. We manually set the starting point of the trajectory from the cluster where the cells collectively possess a maximum stemness score. Estimation of the cellular count of expressed ORs and their mean expression pattern were plotted along the pseudotime.

**Classification of bulk tumor profiles**. Generation of a fine-grained gene expression signature that harbors information about the cellular count of expressed ORs and their individual expression levels from single-cell sequencing datasets was performed as follows. The Expression data matrix (TCGA and single-cell breast cancer datasets) were $\log_2$ transformed ($\log_2$(TPM+1)) and the estimation of the number of expressed ORs per cell was performed based on zFPKM cutoff ($>-3$). We applied this criterion for the estimation of active/non-active ORs in each malignant cell and obtained the binary matrix for downstream clustering (if zFPKM = $<-3$ then 0, otherwise 1). Clustering on the resulting OR binary matrix was performed using the hierarchical clustering method. In all the cases (three independent breast cancer datasets), we obtained multiple clusters representing cell subpopulations with distinct numbers of expressed ORs per cell. Using this information, we computed the OR signatures (each signature represents a different number of expressed ORs per cell) and projected them on tumor samples from TCGA. Single cell-based OR signatures were generated by averaging all cells within a cluster. The projection of the obtained signatures on the TCGA bulk tumor profiles was determined using

$$\text{cosine similarity } (A, B) = \frac{A \cdot B}{||A|| \times ||B||} = \frac{\sum_{i=1}^{n} A_i \times B_i}{\sqrt{\sum_{i=1}^{n} A_i{}^2} \times \sqrt{\sum_{i=1}^{n} B_i{}^2}},$$

where A and B are two independent vectors.

Finally, we segregated the TCGA bulk expression profiles based on hierarchical clustering on the basis of the cosine similarity matrix. Group-specific survival probabilities were estimated using the Kaplan–Meier method. Notably, Kaplan–Meier is a non-parametric estimator of survival probabilities based on patients' longitudinal lifetime data[68].

**Statistics and reproducibility**. Graphical illustrations are plotted with R Stats packages. The P-value cut-off used in this study is 0.05. *, **, ***, and **** in the figures refer to P-values ≤ 0.05, ≤0.01, ≤0.001, and ≤0.0001, respectively. For comparison of the medians of the two distributions, the Mann–Whitney U Test was performed, whereas the Shapiro–Wilk test was used for the correlation.

All the scripts, along with the raw data to reproduce every figure is provided in this link. (https://github.com/the-ahuja-lab/CancerSmell/tree/master/Figures).

**Reporting summary**. Further information on research design is available in the Nature Research Reporting Summary linked to this article.

## Data availability
The single-cell expression matrices (non-normalized) of malignant cells across multiple cancer cell lines and tumor samples were downloaded from CancerSea, a PanTumor single-cell RNAseq database (http://biocc.hrbmu.edu.cn/CancerSEA/home.jsp)[69]. Following are the accession ids of the datasets used in this study: GSE102130, GSE69405, GSE73121, GSE77308, GSE83142, GSE85534, GSE67980, GSE75367, GSE110499, GSE76312, GSE83533, GSE102130, GSE57872, GSE70630, GSE84465, GSE89567, GSE75688, GSE81861, E-MTAB-6149, GSE103322, GSE72056, GSE81383, GSE99330, GSE97681, GSE113660, DRP003981, GSE99305, E-MTAB-6142, GSE99795, DRP001358, GSE81812, GSE68596, GSE81861, GSE85534, GSE76312, GSE98734, GSE65525, ERP020478, GSE51254, GSE57872, GSE102130, GSE80297, GSE81861, DRP003337. Notably, datasets provided in the CancerSea include only those cells which were positive for cellular malignancy. We implemented CancerSmell on those datasets in which the minimum number of malignant single cells was at least 60. Notably, to recheck the authenticity of the downloaded data, we have randomly downloaded a subset of raw files and reanalyzed, and found no discrepancies.

## Code availability
An end-to-end bioinformatics pipeline (CancerSmell) for chemosensory receptor detection in single-cells datasets is provided from the following GitHub (CancerSmell: https://github.com/the-ahuja-lab/CancerSmell) and Zenodo (https://zenodo.org/badge/latestdoi/236710981). An R package enabling OR-based stratification of the patient's

cohort using a single-cell expression-based gene signature (ORsurv: https://github.com/krishan57gupta/ORsurv).

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

## Acknowledgements

The authors would like to thank the IT-HelpDesk team of IIIT-Delhi for providing assistance with the computational resources. We thank all the members of the Ahuja lab for intellectual contributions at various stages of this project. The Ahuja lab is supported by the Ramalingaswami Re-entry Fellowship, are-entry scheme of the Department of Biotechnology (DBT), Ministry of Science & Technology, Govt. of India, and an intra-mural Start-up grant from Indraprastha Institute of Information Technology-Delhi (IIIT-Delhi). The Sengupta lab is funded by the INSPIRE faculty grant from the Department of Science & Technology (DST), India.

## Author contributions

The study was conceived by G.A., Experimental workflows were designed by G.A., D.S., S.K., S.N. and performed by S.K., A.M., K.G., V.S., T.M. and A.G. Illustrations were drafted by G.A. and A.M. G.A. and D.S. wrote the paper. All authors have read and approved the manuscript.

## Competing interests

Debarka Sengupta is an Editorial Board Member for Communications Biology, but was not involved in the editorial review of, nor the decision to publish this article. The authors declare no other financial or non-financial competing interests.
