## [Peer Review File · Communications Biology]

Reviewers' comments:

Reviewer #1 (Remarks to the Author):

In the manuscript entitled "Integrative analysis of single-cell transcriptomes links cellular enrichment of olfactory receptors with cancer cell differentiation and molecular prognosis" by Siddhant Kalra and coworkers performed a detailed analysis of ectopic odorant receptor expression in single cancer cells.

General critique:

The analysis is well done, and with no doubt the results are of high interest not only to the community of cancer genetics and therapy, but also to scientist of other fields. Some of the obtained results have with no doubt the potential to influence the thinking and further research in the field.

Unfortunately, some passages of the manuscript appear written rather hastily, some crucial background information necessary to understand the study is missing, and not all points are adequately referenced. The Methods need to be better introduced and explained. The Discussion is rather short, and the authors should discuss in more detail what was already known about odorant receptors and cancer.

Throughout the manuscript some abbreviations are not spelled out at their first appearance.

Most disturbingly, it is very difficult to keep track of the immense amount of presented data. The authors do not always explain the data in an appropriate way. The Figures (and supplementary Figures) are overloaded with data and are therefore very difficult to read. The Figure legends are not detailed enough to understand what many parts of the Figures show. I suggest reducing data and Figures to essential points that can be explained in appropriate detail in the manuscript text and Figure legends. This would substantially increase readability and understandability of the paper.

Although I'm not a native English speaker, I also feel that syntax and punctuation throughout the manuscript could be slightly improvement.

Specific critique:

Abstract

- The statement that odorant receptors "go unnoticed in cancer" should not be made. A lot of data about odorant receptors expression in cancer already exists.
- TCGA? Please introduce this abbreviation.
- The second to last sentence of the Abstract is not easy to understand. Try to reformulate.

Introduction

- The authors could put more effort in introducing odorant receptors (transduction pathway, transduction channels, etc.), and give a little more information about the fact that they have long been known to be ectopically expressed in many different tissues.
- Ligand-mediated odorant receptor activation in cancer? The authors should give more information about possible ligands. Are these ligands also odorants? Where do these ligands come from? Is this

known?

- "Traditional, bulk-tissue based transcriptomics fails to capture the OR expression signal in tumors, mainly due to the selective expression of these molecules in cancer cells (29,30)." Please better explain what you mean with this sentence.
- "In the present study, we investigated OR activation in single malignant cells leveraging numerous publicly available single-cell RNA sequencing (scRNA-Seq) datasets." You repeat the term "OR activation" throughout the manuscript. Please explain what you mean with odorant receptor activation. I guess you do not mean activation of odorants by ligands/odorants.

3. Material and Methods

- Page 5: Please better introduce and explain the "Cancer Smell computational workflow".

4. Results

- In the first part of the Results you speak about vomeronasal type 2, trace amine-associated receptors, and taste receptors. Please give more details about these receptor families. I suggest introducing them already in the Introduction section. You need to state that there are different (olfactory) receptor families.
- "Trace Amine-Associated Receptors". TAARs is the abbreviation, they are not also known (aka) as TAARs. The same applies to V2Rs.
- Page 9: "Notably, among them, we have observed enrichment of OR genes, in contrast to Trace Amine-Associated Receptors aka TAARs, vomeronasal type-2 receptors aka V2Rs and taste receptors, which is in line with the earlier reports (Figure 1A, C-F; S1B, F; Table1)". Does this statement mean that enrichment of odorant receptor gene enrichment has never been observed in earlier reports?
- Page 10, line 5: "molecules". What do you mean with molecules? Odorant receptors?
- Page 10: What do you mean with the activation status of chemosensory receptors?
- Page 11: "Noteworthy Triple-Negative Breast Cancer (TNBC) cells, being the most aggressive breast cancer subtype, displayed the highest levels of enrichment of OR repertoire (Figure S2E; Table 5) (40)." Why do you add a reference (40) to this result?
- Page 12: Here you start speaking about an epigenetic mechanism behind the simultaneous activation of multiple odorant receptors. I think it is necessary to better explain/ introduce this epigenetic mechanism. What exactly do you mean?
- Page 12: patient BC05. This is the only place where you mention a patient. Which patient? Unclear.

5. Discussion

- Page 15: T1Rs, and T2Rs. This is the first time that this abbreviations appear. Better spell out here.
-

6. Figures

- As already mentioned in my general critique both Figures and supplementary Figures are overloaded with data. Also, the Figure legends are not detailed enough to understand what the Figures show.

Reviewer #2 (Remarks to the Author):

This is overall an interesting paper written with the necessary detail. However, as we noted below, several places require further clarification.

Pg. 5: what is source/citation providing the recommendation for a zFPKM cutoff > -3 ?

Pg. 6: not clear what the following sentence means: "Differential gene expression analysis using a custom-build script, utilizing the Wilcoxon-test for the estimation of statistical significance, if any."

Pg. 7: define and discuss the concept "degree of stemness"

Pg. 7: not clear what "OR frequencies" mean in the following sentence: "In most cases, we have obtained > 2 clusters (groups of malignant cells) segregated based on cellular OR frequencies."

Pg. 8: the authors should add citation and give a brief explanation for Kaplan-Meier estimator

Pg. 8: was the p-value cutoff of 0.05 corrected for multiple hypothesis testing?

Pg. 10: not clear what the following text means: "transcriptional rules mediating OR co-expression are broadly regulated"

Pg. 11: what does "sub-fraction" mean?

Point-By-Point response

Reviewers' comments:

Reviewer #1 (Remarks to the Author):

- In the manuscript entitled “Integrative analysis of single-cell transcriptomes links cellular enrichment of olfactory receptors with cancer cell differentiation and molecular prognosis” by Siddhant Kalra and coworkers performed a detailed analysis of ectopic odorant receptor expression in single cancer cells.

General critique:

- *The analysis is well done, and with no doubt the results are of high interest not only to the community of cancer genetics and therapy, but also to scientist of other fields. Some of the obtained results have with no doubt the potential to influence the thinking and further research in the field.*

We thank the reviewer for appreciating our findings.

- *Unfortunately, some passages of the manuscript appear written rather hastily, some crucial background information necessary to understand the study is missing, and not all points are adequately referenced. The Methods need to be better introduced and explained. The Discussion is rather short, and the authors should discuss in more detail what was already known about odorant receptors and cancer.*

We thank the reviewer for pointing this out. We apologize for these limitations, which we have now addressed in the revised manuscript. In particular, we have now expanded

the introduction section by incorporating more shreds of evidence from the literature. Moreover, we have also elaborated on various sections of the Methods. The discussion has also been substantially extended.

- *Throughout the manuscript some abbreviations are not spelled out at their first appearance.*

We have rectified the issue in the revised manuscript.

- *Most disturbingly, it is very difficult to keep track of the immense amount of presented data. The authors do not always explain the data in an appropriate way. The Figures (and supplementary Figures) are overloaded with data and are therefore very difficult to read. The Figure legends are not detailed enough to understand what many parts of the Figures show.*

We thank the reviewer for pointing this out. We would like to highlight that in the revised manuscript, we have split the few bulky main figures and supplementary figures. Moreover, as per the reviewer's suggestion, we expanded the figure legends by incorporating the relevant details.

- *I suggest reducing data and Figures to essential points that can be explained in appropriate detail in the manuscript text and Figure legends. This would substantially increase readability and understandability of the paper.*

We thank the reviewer for his/her suggestion. We have now distributed the results of bulky figures into new figures. Furthermore, we have also expanded the figure legends by incorporating relevant details.

- Although I'm not a native English speaker, I also feel that syntax and punctuation throughout the manuscript could be slightly improved.

We thank the reviewer for his/her suggestion. We have to highlight that the language of the revised manuscript is now thoroughly checked by an English language expert. Moreover, we have also used “Grammarly” to further ensure accurate language usage.

- *Specific critique:*

Abstract

The statement that odorant receptors “go unnoticed in cancer” should not be made. A lot of data about odorant receptors expression in cancer already exists.

We apologize for this. We have replaced this statement in the revised manuscript.

- TCGA? Please introduce this abbreviation.

We thank the reviewer for pointing this out. This has now been rectified in the revised manuscript.

- The second to the last sentence of the Abstract is not easy to understand. Try to reformulate.

We thank the reviewer for pointing this out. We have made the suggested changes to this sentence in the revised manuscript.

- **Introduction**

The authors could put more effort in introducing odorant receptors (transduction pathway, transduction channels, etc.), and give a little more information about the fact that they have long been known to be ectopically expressed in many different tissues.

We thank the reviewer for pointing this out. We would like to highlight that in the introduction section of the revised manuscript we have added a detailed explanation for

the olfactory receptors, their types, and the downstream signal transduction pathways. Moreover, for the second suggestion about the known ectopic ORs, we have now discussed them in the discussion section of the revised manuscript.

- *Ligand-mediated odorant receptor activation in cancer? The authors should give more information about possible ligands. Are these ligands also odorants? Where do these ligands come from? Is this known?*

We thank the reviewer for pointing this out. In the revised manuscript we have mentioned some of the seminal works in the introduction section of the manuscript. This include:

1. The activation of OR1A2 by *citronellal* in hepatocellular carcinoma.
2. Troenan induced activation of OR51B4 in colorectal cancer cells.
3. Activation of OR51E2 by an endogenous agonist, 19-hydroxyandrostenedione.
4. Activation of OR51E2 in vertical-growth phase melanoma cells by β -ionone.
5. The activation of OR2J3 by helional in non-small-cell lung cancer.

In the majority of cases, the known ligands are mostly odorant molecules, except in the case of OR51E2, whose one of the known endogenous agonists, 19-hydroxyandrostenedione, is known. Moreover, we assume that the cancer-specific metabolic intermediates could be the potential endogenous agonists for these cancer-specific ectopic ORs identified in this study. We have now incorporated these assumptions in the discussion section of the manuscript.

- *“Traditional, bulk-tissue based transcriptomics fails to capture the OR expression signal in tumors, mainly due to the selective expression of these molecules in cancer cells (29,30).” Please better explain what you mean with this sentence.*

We thank the reviewer for pointing this out. In the revised manuscript, we have rephrased this sentence.

- “In the present study, we investigated OR activation in single malignant cells leveraging numerous publicly available single-cell RNA sequencing (scRNA-Seq) datasets.” You repeat the term “OR activation” throughout the manuscript. Please explain what you mean with odorant receptor activation. I guess you do not mean activation of odorants by ligands/odorants.

We apologize for this. We have to highlight that “OR activation” refers to the reliable presence of the OR transcripts within the cell, not the ligand-mediated OR activation. To avoid confusion, in the revised manuscript we have replaced the word “activation”.

- *3. Material and Methods*

Page 5: Please better introduce and explain the “Cancer Smell computational workflow”.

We thank the reviewer for his/her comment. A detailed explanation of the workflow has been added to the revised manuscript. Notably, we have also provided a GitHub link that contains a step-by-step explanation of all the codes used in this study.

GitHub link: <https://github.com/the-ahuja-lab/CancerSmell>

- *4. Results*

In the first part of the Results you speak about vomeronasal type 2, trace amine-associated receptors, and taste receptors. Please give more details about these receptor families. I suggest introducing them already in the Introduction section. You need to state that there are different (olfactory) receptor families.

We thank the reviewer for his/her suggestion. We have to highlight that in the introduction section of the revised manuscript, we have added the suggested information.

- *“Trace Amine-Associated Receptors”. TAARs is the abbreviation, they are not also known (aka) as TAARs. The same applies to V2Rs.*

We thank the reviewer for pointing this out. We have corrected this in the revised manuscript.

- *Page 9: “Notably, among them, we have observed enrichment of OR genes, in contrast to Trace Amine-Associated Receptors aka TAARs, vomeronasal type-2 receptors aka V2Rs and taste receptors, which is in line with the earlier reports (Figure 1A, C-F; S1B, F; Table1)”. Does this statement mean that enrichment of odorant receptor gene enrichment has never been observed in earlier reports?*

We apologize for this confusion. What we meant was that among all the chemosensory receptor gene families i.e. ORs, TAARs, V2Rs, and taste receptors, the comparative abundance of ORs to be ectopically expressed in the cancer state is much higher. This has already been reported in the literature by multiple groups using bulk transcriptomics or microarray technologies. In our study, despite scanning the expression prevalence of all the chemosensory receptor gene families in the given cancer cells, we have also observed selective enrichment of ORs. Our results are largely similar to the earlier findings where ORs were found to be *comparatively* enriched in the cancer cells, in contrast to other chemosensory receptors. Notably, we have evaluated 22 tumor types and collectively comprising 42,529 malignant cells and we could merely find *VN1R1* and *VN1R2* to be expressed only in skin cutaneous melanoma (SKCM) while *TAS1R3* was found to be expressed only in Cervical Squamous Cell Carcinoma (CESC). These results indicate the ectopic expression bias of ORs in the cancer state.

- *Page 10, line 5: “molecules”. What do you mean with molecules? Odorant receptors?*

We apologize for this. In the revised manuscript we have replaced the word “molecules” by ‘receptor’.

- *Page 10: What do you mean with the activation status of chemosensory receptors?*

We thank the reviewer for pointing this out. We have to highlight that the term “*activation status of chemosensory receptors*” referred to “the presence of reliable levels of chemosensory receptor transcripts” as inferred by the zFPKM algorithm. Therefore, to avoid confusion, we have replaced the word *activation* in the revised manuscript.

- *Page 11: “Noteworthy Triple-Negative Breast Cancer (TNBC) cells, being the most aggressive breast cancer subtype, displayed the highest levels of enrichment of OR repertoire (Figure S2E; Table 5) (40).” Why do you add a reference (40) to this result?*

We apologize for this mistake. The reference was to justify the statement that Triple Negative Breast Cancer is the most aggressive form of Cancer. The reference was placed in the wrong position. The second part of the statement “displayed the highest level of enrichment of OR repertoire” was the inference from our results which has been depicted in Figure S3E (new). We have now added the reference at the right place in the revised manuscript.

- *Page 12: Here you start speaking about an epigenetic mechanism behind the simultaneous activation of multiple odorant receptors. I think it is necessary to better explain/ introduce this epigenetic mechanism. What exactly do you mean?*

We thank the reviewer for this suggestion. We would like to highlight that the emergence of multiple ORs per neuron during early development is speculated to be epigenetically regulated by the authors (Hanchate et. al. Science 2015). Moreover, so far there are no concrete reports which have discussed the molecular basis of such events. We have added the information about the epigenetic modifications and their role

in regulating the number of expressed ORs per cell in the results and discussion section of the revised manuscript.

- *Page 12: patient BC05. This is the only place where you mention a patient. Which patient? Unclear.*

We thank the reviewer for pointing this out. We have to mention that the supplementary table 7 contains information of 11 patients (excluding BC05), along with their IDs from which the samples were collected. We have now described the patient information in the revised manuscript.

- *5. Discussion*
Page 15: T1Rs, and T2Rs. This is the first time that these abbreviations appear. Better spell out here.

We thank the reviewer for this suggestion. We have now rectified these.

6. Figures

- *As already mentioned in my general critique both Figures and supplementary Figures are overloaded with data. Also, the Figure legends are not detailed enough to understand what the Figures show.*

We would like to highlight that in the revised manuscript, we have split the bulky main figures and supplementary figures. Moreover, as per the reviewer's suggestion, we expanded the figure legends by incorporating the relevant details.

#####

#

Reviewer #2 (Remarks to the Author):

- *This is overall an interesting paper written with the necessary detail. However, as we noted below, several places require further clarification.*

We thank the reviewer for his/her comment.

- *Pg. 5: what is source/citation providing the recommendation for a zFPKM cutoff >-3?*

We apologize for this. We have now added the citation in the revised manuscript.

- *Pg. 6: not clear what the following sentence means: "Differential gene expression analysis using a custom-build script, utilizing the Wilcox-test for the estimation of statistical significance, if any."*

We thank the reviewer for pointing this out. We have rewritten this sentence in the revised manuscript. Following sentence is the replacement in the revised manuscript:

"To identify the differentially expressed genes between the cells harboring high and low numbers of expressed ORs, we first calculated the median of the OR count across all cells. Next, we segregated the cell population (cluster-wise) based on the median value. Last we calculated the differential gene expression analysis using the Wilcox-test. Only statistically significant genes were selected for the functional enrichment analysis (fold change cutoff +-4; corrected p-value < 0.05) using Metascape (<http://metascape.org>)."

- *Pg. 7: define and discuss the concept "degree of stemness"*

We thank the reviewer for pointing this out. We would like to highlight that we have added this information in the revised manuscript.

- *Pg. 7: not clear what "OR frequencies" mean in the following sentence: "In most cases, we have obtained > 2 clusters (groups of malignant cells) segregated based on cellular OR frequencies."*

In order to identify the signatures representing the number of expressed ORs per cells (cellular OR frequencies), we first performed the hierarchical clustering of all the malignant cells. In all the cases (three independent breast cancer datasets), we obtained multiple clusters representing cell populations with distinct numbers of expressed ORs per cell. Using this information, we computed the OR-centric signatures (each signature representing a differential number of expressed ORs per cell) and projected on bulk TCGA tumor samples. We have now added this information in the revised manuscript.

- *Pg. 8: the authors should add citation and give a brief explanation for Kaplan-Meier estimator*

We thank the reviewer for pointing this out. We have added the citation and a brief explanation of the Kaplan-Meier estimator in the revised manuscript.

- *Pg. 8: was the p-value cutoff of 0.05 corrected for multiple hypothesis testing?*

Yes, the p-values were multiple-test corrected using the Bonferroni correction. This information has now been added in the revised manuscript.

- *Pg. 10: not clear what the following text means: "transcriptional rules mediating OR co-expression are broadly regulated"*

The inspection of OR expression at the single-cell resolution revealed combinatorial expressions of ORs in malignant cells. We asked if there is a co-expression bias for any

two ectopic ORs, which could reflect the co-transcriptional regulation. For this, we took the multiple OR pairs and traced their co-expression across all malignant cell types. Our results suggest that the selection of the expressed ORs within a single cell is stochastic in nature. Moreover, these findings were further strengthened by the co-expression analysis where a relatively poor correlation was observed between two highly, albeit co-expressed ORs.

- *Pg. 11: what does "sub-fraction" mean?*

A total of 53 distinct ORs were identified to be expressed in the malignant BRCA dataset. Out of those, 14 were also detected in the normal breast tissue. These 14 olfactory receptors are referred to as sub-fraction. This has now been clarified in the manuscript.

REVIEWERS' COMMENTS:

Reviewer #1 (Remarks to the Author):

The authors have answered in detail all my questions and have addressed my points. The revised version of the manuscript has been substantially revised and improved. I would like to commend the authors for their effort.

I have no additional points.

Reviewer #2 (Remarks to the Author):

The authors have addressed our comments on the first version. The manuscript is clearer and more navigable.

We have a major comment at this point. The authors' concluding remark "our comprehensive analysis positions ORs at the cross-road of tumor cell differentiation and molecular prognosis of cancer" is thought-provoking. But cancer is known to be heterogeneous in so many aspects! There are many other molecular phenotypes "at the cross-road of tumor cell differentiation and molecular prognosis of cancer". At least in the Discussion section, the authors should comment on whether/how ORs are unique among all these molecular phenotypes at the cross-road. It is fine if the question cannot be answered right away, but what (data/method) would the authors need to answer this question?

Point-By-Point response

Reviewer #1 (Remarks to the Author):

The authors have answered in detail all my questions and have addressed my points. The revised version of the manuscript has been substantially revised and improved. I would like to commend the authors for their effort.

I have no additional points.

We thank the reviewer 1 for appreciating our work.

Reviewer #2 (Remarks to the Author):

The authors have addressed our comments on the first version. The manuscript is clearer and more navigable.

We have a major comment at this point. The authors' concluding remark "our comprehensive analysis positions ORs at the cross-road of tumor cell differentiation and molecular prognosis of cancer" is thought-provoking. But cancer is known to be heterogeneous in so many aspects! There are many other molecular phenotypes "at the cross-road of tumor cell differentiation and molecular prognosis of cancer". At least in the Discussion section, the authors should comment on whether/how ORs are unique among all these molecular phenotypes at the cross-road. It is fine if the question cannot be answered right away, but what (data/method) would the authors need to answer this question?

We thank the reviewer 2 for appreciating our work. We agree with the reviewer's suggestion and therefore added a few sentences regarding the uniqueness of ORs among all the known molecular phenotypes at the cross-road of tumor-cell differentiation and molecular prognosis in the discussion section of the revised manuscript.